# Fixing the Bethe Approximation:
# How Structural Modifications in a Graph Improve Belief Propagation

Harald Leisenberger[1]            Franz Pernkopf[1]            Christian Knoll[1]

[1]Signal Processing and Speech Communication Laboratory, Graz University of Technology, Austria

## Abstract

Belief propagation is an iterative method for inference in probabilistic graphical models. Its well-known relationship to a classical concept from statistical physics, the Bethe free energy, puts it on a solid theoretical foundation. If belief propagation fails to approximate the marginals, then this is often due to a failure of the Bethe approximation. In this work, we show how modifications in a graphical model can be a great remedy for fixing the Bethe approximation. Specifically, we analyze how the removal of edges influences and improves belief propagation, and demonstrate that this positive effect is particularly distinct for dense graphs.

## 1 INTRODUCTION

Message passing algorithms are an effective method for approximate inference in probabilistic graphical models [Koller and Friedman, 2009]. Although they often perform well in practice, there are only few guarantees about their theoretical behavior. A remarkable milestone was the discovery of a direct connection between message passing algorithms and concepts of statistical mechanics [Yedidia et al., 2001], perhaps the most famous being the relationship between belief propagation (BP) [Pearl, 1988] and the so-called Bethe free energy [Bethe, 1935, Peierls, 1936].

One favorable property of BP is its exactness on trees. On loopy graphs, however, it frequently suffers from two major issues: first, it may fail to converge to a fixed point and thus to find reasonable estimates of the marginals. Second, the fixed points themselves may induce bad estimates of the marginals, in which case even convergence would not help.

To solve the first issue, various techniques have been developed that can improve the convergence behavior of BP; e.g., one can damp the message updates [Murphy et al., 1999]

or utilize elaborate scheduling schemes [Elidan et al., 2006, Sutton and McCallum, 2007, Knoll et al., 2015, Aksenov et al., 2020]. Moreover, it depends on the properties of the graphical model [Tatikonda and Jordan, 2002, Ihler et al., 2005, Mooij and Kappen, 2007] and on the initialization of the messages [Koehler, 2019, Knoll et al., 2021, Leisenberger et al., 2021] if and to which fixed point BP converges.

The second issue might be even harder to overcome, as it is inherently linked to a failure of the Bethe approximation [Weller et al., 2014]. The detrimental influence of loops can make the Bethe free energy non-convex and cause its local minima – and thus BP fixed points – to be far away from the exact marginals. Enhanced variants of free energy approximations [Yedidia et al., 2005] or loop corrections [Mooij et al., 2007] are prudent alternatives, that improve the accuracy but also increase the complexity.

In this work, we follow a different path: we aim to improve the approximation quality of the Bethe free energy itself. To address this problem, we modify the structure of the graphical model and show how this transforms the Bethe free energy in a way that moves its local minima closer to the exact marginals. In particular, we analyze the effect of removing individual edges from the graph. This loop-breaking approach enforces a 'reconvexification' of the Bethe free energy and therefore not only improves the accuracy of fixed points, but also the convergence behavior of BP.

We make a series of interesting theoretical contributions that arise from analyzing the behavior of the Bethe free energy on a 'small scale'. More precisely, we introduce a measure for the discrepancy between two different representations of the Bethe free energy, each induced by a different graphical model, and then utilize this tool to relate variations in the Bethe free energy to the characteristics of the model and the behavior of BP. Theoretically and experimentally, we address the following questions: (i) How does edge removal influence the estimated marginals? (ii) How does edge removal influence the estimated partition function? (iii) Which and how many edges – if at all – should we remove?

*Accepted for the 38th Conference on Uncertainty in Artificial Intelligence* (UAI 2022).

The structure of this paper is as follows: Sec. 2 summarizes all relevant background on graphical models, BP, and the Bethe approximation. Sec. 3 contains a detailed theoretical analysis and the main results. We experimentally validate our findings in Sec. 4 and conclude the paper in Sec. 5.

## 2 BACKGROUND

This introductory section provides a compact overview of the topics that we deal with: probabilistic graphical models, belief propagation, and the Bethe approximation. We further introduce the Ising model and discuss related work.

### 2.1 PROBABILISTIC GRAPHICAL MODELS

We consider an undirected graph $\mathcal{G} = (\mathbf{X}, \mathbf{E})$, where $\mathbf{X} = \{1, \ldots, N\}$ is a set of nodes and $\mathbf{E} \subseteq \{(i,j) : i,j \in \mathbf{X}\}$ is a set of edges. An edge connects two nodes if $(i,j) \in \mathbf{E}$. Note that we assume all edges to be undirected and hence $(i,j) = (j,i)$ for all pairs of connected nodes; specifically we do not count edges twice. Furthermore, $\mathcal{N}(i)$ denotes the neighborhood of node $i$ (i.e., the set of nodes that are connected to $i$) and $d_i := |\mathcal{N}(i)|$ denotes the degree of $i$.

Let $X_1, \ldots, X_N$ be random variables (RVs) with state spaces $\mathcal{X}_1, \ldots, \mathcal{X}_N$. A probabilistic graphical model (PGM) represents a joint probability distribution $P_{\mathbf{X}}(\boldsymbol{x})$ over the RVs, where each node represents a RV[1] and edges indicate statistical dependencies between RVs. Formally, a PGM is a pair $(\mathcal{G}, \boldsymbol{\Phi})$ that associates a set of potential functions (or potentials) $\boldsymbol{\Phi} = \{\Phi_1(\boldsymbol{x_1}), \ldots, \Phi_K(\boldsymbol{x_K})\}$ with the graph $\mathcal{G}$ that are defined over joint realizations of subsets $\boldsymbol{X_1}, \ldots, \boldsymbol{X_K} \subseteq \mathbf{X}$. We shall focus on the class of binary pairwise models[2] that satisfy the following two assumptions: first, each RV has two states, i.e., $\mathcal{X}_i = \mathcal{X} = \{+1, -1\}$. Second, the potentials are defined over either one (singleton potentials $\Phi_i(x_i)$) or two (pairwise potentials $\Phi_{ij}(x_i, x_j)$) RVs. Then the joint distribution factorizes as

$$P_{\mathbf{X}}(\boldsymbol{x}) = \frac{1}{\mathcal{Z}} \prod_{(i,j)\in\mathbf{E}} \Phi_{ij}(x_i, x_j) \prod_{i=1}^{N} \Phi_i(x_i), \quad (1)$$

where $\mathcal{Z}$ is the normalization constant or partition function.

We consider the Ising model, whose potentials have the form $\Phi_{ij}(x_i, x_j) = \exp(J_{ij}x_i x_j)$ and $\Phi_i(x_i) = \exp(\theta_i x_i)$ with $J_{ij} \in \mathbb{R}$ being the coupling strength of edge $(i,j)$ and $\theta_i \in \mathbb{R}$ being the local field of node $i$. We further call an edge $(i,j)$ *attractive* if $J_{ij} > 0$, and *repulsive* if $J_{ij} < 0$.

Then (1) takes the exponential form

$$P_{\mathbf{X}}(\boldsymbol{x}) = \frac{1}{\mathcal{Z}} \exp(-E(\boldsymbol{x})) \quad (2)$$

with $E(\boldsymbol{x}) := \sum_{(i,j)\in\mathbf{E}} J_{ij}x_i x_j + \sum_{i=1}^{N} \theta_i x_i$ being the state energy.[3]

Finally, following the terminology in Knoll et al. [2021], we specify three types of models: *unidirectional* models do only contain attractive edges and all variables are biased towards the same state (i.e., either $\theta_i \leq 0$ or $\theta_i \geq 0$); *attractive* models do only contain attractive edges, but there may be local fields that differ in sign; *general* models may contain both attractive and repulsive edges. Note that, by definition, attractive models include unidirectional models, while general models include both unidirectional and attractive models.

### 2.2 BELIEF PROPAGATION

In this work, we consider two problems: first, the computation of marginal distributions where our specific interest lies in singleton marginals $P_{X_i}(x_i)$; and second, the computation of the partition function.[4] It is well known that an exact computation of these quantities is intractable [Valiant, 1979, Cooper, 1990] and even the approximation of the marginals to a certain precision is NP-hard [Dagum and Luby, 1993].

Belief propagation (BP) approximates the marginals by iteratively exchanging local statistical information between nodes in form of 'messages'. This process is governed by the recursive message update equations

$$\mu_{ij}^{(n+1)}(x_j) \propto \sum_{x_i \in \mathcal{X}} \Phi_{ij}(x_i, x_j)\Phi_i(x_i) \prod_{k \in N(i)\backslash j} \mu_{ki}^{(n)}(x_k), \quad (3)$$

where the superscript $(n)$ refers to the current iteration and the subscript $ij$ refers to the direction in which a message is sent (e.g., node $i$ sends $\mu_{ij}$ to node $j$). In principle, one can approximate the singleton marginals at any iteration, by multiplying all incoming messages with the local potential:

$$\tilde{P}_i^{(n)}(x_i) \propto \frac{1}{Z_i} \Phi_i(x_i) \prod_{k \in \mathcal{N}(i)} \mu_{ki}^{(n)}(x_k) \quad (4)$$

Generally, however, marginal estimates are considered to be more accurate when they are obtained from a BP fixed point [Murphy et al., 1999, Knoll et al., 2017]; more precisely, BP has converged to a fixed point $\mu_{ij}^{\circ}$, whenever an update of the form (3) does not alter the message values anymore (that is $\mu_{ij}^{(n+1)} = \mu_{ij}^{(n)}$ for all $(i,j)$).

---

[1]Due to the one-to-one correspondence between variables $X_i$ and nodes $i$, we shall not rigorously distinguish between them; e.g., we often write $P_i(x_i)$ instead of $P_{X_i}(x_i)$.

[2]A wide range of models can be equivalently transformed into binary pairwise models, although this may increase the state space considerably [Weiss, 2000, Eaton and Ghahramani, 2013].

[3]This parameterization does often facilitate the theoretical analysis as it associates a unique parameter vector – consisting of all $J_{ij}$ and $\theta_i$ – with each distribution (a so-called minimal representation, Wainwright et al. [2008]).

[4]Actually, these two problems are closely related as marginals can be expressed as a ratio of sub-partition functions [Weller and Jebara, 2014b].

## 2.3 BETHE APPROXIMATION

It is often useful to formulate marginal inference in terms of a variational problem. For this purpose, we consider some trial distribution $Q_{\mathbf{X}}(\boldsymbol{x})$ and define the Gibbs free energy as

$$\mathcal{F}(Q_{\mathbf{X}}) = \mathcal{U}(Q_{\mathbf{X}}) - \mathcal{S}(Q_{\mathbf{X}}) \qquad (5)$$

with $\mathcal{U} = \mathbb{E}_Q[E(\mathbf{X})]$ being the average energy of the model and $\mathcal{S} = -\sum_{\boldsymbol{x} \in \mathcal{X}^N} Q_{\mathbf{X}}(\boldsymbol{x}) \log Q_{\mathbf{X}}(\boldsymbol{x})$ being the entropy of $Q_{\mathbf{X}}(\boldsymbol{x})$. Let us further define the marginal polytope $\mathbb{M}$ as the set of all valid probability distributions over $\mathbf{X}$ (i.e., that satisfy all global and local marginalization and normalization constraints). If one minimizes $\mathcal{F}$ over $\mathbb{M}$, then one recovers the true distribution $P_{\mathbf{X}}(\boldsymbol{x})$ with the negative log-partition function $-\log(\mathcal{Z})$ as the functional value at the global minimum, i.e., $-\log(\mathcal{Z}) = \min_{\mathbb{M}} \mathcal{F}(Q_{\mathbf{X}}) = \mathcal{F}(P_{\mathbf{X}})$.[5]

Two aspects, however, render the minimiziation of the Gibbs free energy intractable: first, the definition of the marginal polytope by exponentially many constraints; second, the evaluation of the entropy that requires summing over exponentially many terms. The Bethe free energy approximation addresses these two issues as follows: first, it relaxes the marginal polytope $\mathbb{M}$ to the local polytope $\mathbb{L}$ that involves only local marginalization and normalization constraints of the pairwise and singleton 'pseudo-marginals' $\tilde{P}_{ij}$ and $\tilde{P}_i$:

$$\mathbb{L} = \{\tilde{P}_{ij}, \tilde{P}_i : \sum_{x_j} \tilde{P}_{ij}(x_i, x_j) = \tilde{P}_i(x_i),$$
$$\sum_{x_i, x_j} \tilde{P}_{ij}(x_i, x_j) = 1, \sum_{x_i} \tilde{P}_i(x_i) = 1; \quad (6)$$
$$(i, j) \in \mathbf{E}, i \in \mathbf{X}\}.$$

Second, it replaces the entropy $\mathcal{S}$ by an accordingly weighted sum of entropy contributions from edges and nodes. More concretely, the Bethe free energy is defined as $\mathcal{F}_B = \mathcal{U}_B - \mathcal{S}_B$ where the Bethe average energy is

$$\mathcal{U}_B = -\sum_{(i,j) \in \mathbf{E}} \sum_{x_i, x_j \in \mathcal{X}} \tilde{P}_{ij}(x_i, x_j) \log \Phi_{ij}(x_i, x_j)$$
$$-\sum_{i=1}^n \sum_{x_i \in \mathcal{X}} \tilde{P}_i(x_i) \log \Phi_i(x_i), \qquad (7)$$

and the Bethe entropy is

$$\mathcal{S}_B = -\sum_{(i,j) \in \mathbf{E}} \sum_{x_i, x_j \in \mathcal{X}} \tilde{P}_{ij}(x_i, x_j) \log \tilde{P}_{ij}(x_i, x_j)$$
$$+\sum_{i=1}^n (d_i - 1) \sum_{x_i \in \mathcal{X}} \tilde{P}_i(x_i) \log \tilde{P}_i(x_i). \qquad (8)$$

---

[5]For more details on variational inference in graphical models and the marginal polytope, we refer the reader to Wainwright et al. [2008], Mezard and Montanari [2009].

While $\mathcal{U}_B$ equals the true average energy $\mathcal{U}$ in the exact marginals, $\mathcal{S}_B$ is only an approximation to the true entropy $\mathcal{S}$ – unless the graph is a tree [Yedidia et al., 2005]. To obtain locally consistent approximations to the exact marginals, one usually aims to minimize $\mathcal{F}_B$ over $\mathbb{L}$. Also, the so-called Bethe partition function $\mathcal{Z}_\mathcal{B}$, that is implicitly defined by $-\log(\mathcal{Z}_\mathcal{B}) = \min_{\mathbb{L}} \mathcal{F}_B$, provides an estimation of $\mathcal{Z}$.

Binary variables allow for a particularly simple description of the local polytope. Following the notation of Welling and Teh [2001], Weller and Jebara [2013], we define the pseudo-marginal distribution of $X_i$ by $\tilde{P}_i(X_i = +1) = q_i$ (implying $\tilde{P}_i(X_i = -1) = 1 - q_i$) and, for any pair of connected nodes, we denote the joint pseudo-marginal probability $\tilde{P}_{ij}(X_i = +1, X_j = +1)$ by $\xi_{ij}$. Then the local marginalization and normalization constraints induce a full specification of the joint probability table between $X_i$ and $X_j$ in terms of the three parameters $q_i, q_j, \xi_{ij}$ (Tab. 1).

Table 1: Variational joint probability table for two binary variables $X_i$ and $X_j$.

| $\tilde{P}_{ij}(X_i, X_j)$ | $X_j = +1$ | $X_j = -1$ | |
|---|---|---|---|
| $X_i = +1$ | $\xi_{ij}$ | $q_i - \xi_{ij}$ | $q_i$ |
| $X_i = -1$ | $q_j - \xi_{ij}$ | $1 + \xi_{ij} - q_i - q_j$ | $1 - q_i$ |
| | $q_j$ | $1 - q_j$ | |

If we assume that all probabilities are strictly positive, then $\xi_{ij}$ is bounded by

$$\max(0, q_i + q_j - 1) < \xi_{ij} < \min(q_i, q_j). \qquad (9)$$

Inserting singleton and pairwise pseudo-marginals from Table 1 together with the Ising potentials from Sec. 2.1 into (7) and (8), the Bethe free energy becomes

$$\mathcal{F}_B = -\sum_{(i,j) \in \mathbf{E}} (1 + 2(2\xi_{ij} - q_i - q_j)) J_{ij}$$
$$+ \sum_{i=1}^n (1 - 2q_i) \theta_i \qquad (10)$$
$$- \sum_{(i,j) \in \mathbf{E}} \mathcal{S}_{ij} + \sum_{i=1}^n (d_i - 1) \mathcal{S}_i,$$

where the pairwise entropies are

$$\mathcal{S}_{ij} = -\xi_{ij} \log \xi_{ij}$$
$$- (1 + \xi_{ij} - q_i - q_j) \log(1 + \xi_{ij} - q_i - q_j)$$
$$- (q_i - \xi_{ij}) \log(q_i - \xi_{ij}) \qquad (11)$$
$$- (q_j - \xi_{ij}) \log(q_j - \xi_{ij})$$

and the local entropies are

$$\mathcal{S}_i = -q_i \log q_i - (1 - q_i) \log(1 - q_i). \qquad (12)$$

Then, with $(\boldsymbol{q}; \boldsymbol{\xi})$ being the vector that contains all $q_i$ and $\xi_{ij}$, the local polytope takes the simplified form

$$\mathbb{L} = \{(\boldsymbol{q}; \boldsymbol{\xi}) \in \mathbb{R}^{|\mathbf{X}|+|\mathbf{E}|} : 0 < q_i < 1, i \in \mathbf{X};$$
$$\max(0, q_i + q_j - 1) < \xi_{ij} < \min(q_i, q_j), (i,j) \in \mathbf{E}\}. \quad (13)$$

To further facilitate the task of minimizing $\mathcal{F}_B$ over $\mathbb{L}$, Welling and Teh [2001] have derived necessary conditions for points $(\boldsymbol{q}; \boldsymbol{\xi})$ of $\mathbb{L}$ to be located at local minima of $\mathcal{F}_B$. By setting the partial derivative $\frac{\partial}{\partial \xi_{ij}} \mathcal{F}_B$ for an arbitrary edge to zero, they proved that the resulting quadratic equation

$$\alpha_{ij}\xi_{ij}^2 - (1 + \alpha_{ij}(q_i + q_j))\xi_{ij} + (1 + \alpha_{ij})q_i q_j = 0, \quad (14)$$

$$\text{where} \quad \alpha_{ij} = e^{4J_{ij}} - 1, \quad (15)$$

has a unique valid (i.e., inside the bounds (9)) solution

$$\xi_{ij}^*(q_i, q_j) = \frac{1}{2\alpha_{ij}} \Big( (1 + \alpha_{ij}(q_i + q_j))$$
$$- \sqrt{(1 + \alpha_{ij}(q_i + q_j))^2 - 4\alpha_{ij}(1 + \alpha_{ij})q_i q_j} \Big). \quad (16)$$

This means that for each edge $(i, j)$, the only $\xi_{ij}^*(q_i, q_j)$, that can be located at a stationary point of $\mathcal{F}_B$, depends directly on $q_i$ and $q_j$ and may therefore be inserted in the definition of $\mathcal{F}_B$ (10). This is advantageous for two reasons: first, it considerably reduces the number of independent variables that are involved in optimizing $\mathcal{F}_B$ (i.e., from $|\mathbf{X}| + |\mathbf{E}|$ to $|\mathbf{X}|$); second, it simplifies the shape of the domain, as $\mathcal{F}_B$ is now defined on a box-constrained domain, the *Bethe box*

$$\mathbb{B} = \{\boldsymbol{q} \in \mathbb{R}^{|\mathbf{X}|} : 0 < q_i < 1, i \in \mathbf{X}\}. \quad (17)$$

In this work, we do always refer to the Bethe free energy by $\mathcal{F}_B$, be it defined over the local polytope or the Bethe box.

## 2.4 RELATED WORK

**Belief propagation and the Bethe free energy.** Since the seminal work of Yedidia et al. [2001], it is well known that fixed points of BP correspond one-to-one to stationary points of the Bethe free energy; moreover, stable fixed points of BP must always be associated to local minima of the Bethe free energy [Heskes, 2003].[6] Consequently, one can try to overcome the convergence issue of BP by minimizing the Bethe free directly. To solve the problem, Welling and Teh [2001], Shin [2012] have derived gradient-based algorithms; Yuille [2002], Heskes [2006] have proposed provably convergent double-loop algorithms.

**Variational free energy approximations.** Yedidia et al. [2005] have shown that BP is only a special case of a general class of message passing algorithms, the generalized

belief propagation (GBP). Likewise, fixed points of these algorithms correspond to stationary points of the so-called Kikuchi free energies that try to approximate the true entropy by a sum over entropy contributions from larger node clusters [Kikuchi, 1951, Pelizzola, 2005]. In practice, many of these methods can be prohibitively slow and may suffer in the same way as BP from non-convexity of the particular free energy approximation; i.e., they may – if at all – converge to suboptimal minima. This inspired various researchers to design free energy approximations that are convex [Wainwright et al., 2005, Globerson and Jaakkola, 2007b], some of which are related to convergent message passing algorithms [Kolmogorov and Wainwright, 2006, Globerson and Jaakkola, 2007a, Hazan and Shashua, 2008, Meltzer et al., 2009, Jancsary and Matz, 2011].

**Theoretical work on the Bethe approximation.** The Bethe approximation proves often to be superior to other methods in terms of a tradeoff between efficiency and accuracy [Meshi et al., 2009]. Its theoretical properties have therefore been intensely studied: Heskes [2004], Pakzad and Anantharam [2005] derived conditions for the convexity of the Bethe free energy. Chertkov and Chernyak [2006] formulated the so-called loop series expansion that directly relates the Bethe partition function to the true partition function. Others have found interesting connections between the Bethe approximation and classical graph theory [Watanabe and Fukumizu, 2009, Vontobel, 2013]. Moreover, Weller and Jebara [2014a] derived an FPTAS [7] to approximate the Bethe partition function in attractive models.

**Graphical model approximation.** Another line of research, that is in some sense complementary to variational inference, tries to approximate the graphical model itself. The classical Chow-Liu algorithm [Chow and Liu, 1968] finds a spanning tree such that the Kullback-Leibler (KL) divergence between the original distribution and the induced tree distribution is minimal. Furthermore, two different techniques have been applied to reduce the complexity of exact inference in a graphical model: first, the 'annihilation' of small probabilities that are below a certain treshold [Jensen and Andersen, 1990]; second, the deletion of one or more edges from the model (not necessarily until a spanning tree is reached). Due to its empirical success, the second method deserves special attention: Kjaerulff [1994] carefully selected edges whose removal decreases the treewidth of a graph. van Engelen [1997] studied how the removal of edges in a directed graph influences the KL divergence. Choi and Darwiche [2006] showed that a particular class of GBP, the so-called join graph propagation [Dechter et al., 2002], can be equivalently cast in terms of a procedure that consecutively deletes and recovers edges. In the past, these methods were primarily applied to perform exact inference in the approximated model. For large graphs, this does often remain a hard computational challenge.

---

[6]On the other hand, there may exist minima of the Bethe free energy that are related to unstable BP fixed points [Mooij and Kappen, 2005, Knoll et al., 2017].

---

[7]Fully polynomial-time approximation scheme.

# 3 THEORETICAL ANALYSIS

We shall now devote our attention to the central topic of this work: how removing edges from a graphical model influences the behavior of BP. While the accuracy of the exact marginals degrades if one approximates a model by a sparser one [van Engelen, 1997], one might expect a similar behavior for the marginals estimated by BP. We show, that the opposite is the case: sparsifying the graph does often significantly improve the marginal accuracy of BP. The quality of the estimated partition function, however, tends to degrade by deviating from the original model.

In this section, we explain the second of these phenomena theoretically. We further analyze the role of an 'optimal' edge to be removed and relate this problem to the Bethe free energy. In particular, we prove an inherent relationship between global error measures on the Bethe free energy and the coupling strength of the edges. Our detailed analysis of the Bethe free energy on a 'small scale' extends the work of Welling and Teh [2001], Weller and Jebara [2013], Weller et al. [2014] and leads to better understanding of BP and the Bethe approximation in general.

## 3.1 PROBLEM SPECIFICATION

We briefly clarify the problem to be considered. Let $(\mathcal{G}, \mathbf{\Phi})$ be a PGM and let $(\mathcal{G}', \mathbf{\Phi}')$ be a second PGM that is obtained by removing a set of edges $\tilde{\mathbf{E}}$ (and the associated pairwise potentials) from the original model.[8] Let $\mathcal{P} := \{P_i : i \in \mathbf{X}\}$ be the set of exact (singleton) marginals on $(\mathcal{G}, \mathbf{\Phi})$ and $\mathcal{Z}$ be the partition function. Assume that we run BP on both models and obtain pseudo-marginals $\tilde{\mathcal{P}} := \{\tilde{P}_i : i \in \mathbf{X}\}$ on $(\mathcal{G}, \mathbf{\Phi})$ resp. $\tilde{\mathcal{P}}' := \{\tilde{P}_i' : i \in \mathbf{X}\}$ on $(\mathcal{G}', \mathbf{\Phi}')$, together with partition function estimates $\tilde{\mathcal{Z}}$ resp. $\tilde{\mathcal{Z}}'$.[9] Then we are interested in comparing the following quantities: first, the $l^1$-errors $||\mathcal{P}_\mathbf{X} - \tilde{\mathcal{P}}_\mathbf{X}||_{l^1}$ and $||\mathcal{P}_\mathbf{X} - \tilde{\mathcal{P}}_\mathbf{X}'||_{l^1}$; and second, the absolute errors $|\log \mathcal{Z} - \log \tilde{\mathcal{Z}}|$ and $|\log \mathcal{Z} - \log \tilde{\mathcal{Z}}'|$.

Ideally, we would like to remove a (possibly empty) set of edges, such that the induced errors $||P_\mathbf{X} - \tilde{P}_\mathbf{X}'||_{l^1}$ and $|\log \mathcal{Z} - \log \tilde{\mathcal{Z}}'|$ become minimal over all subsets $\tilde{\mathbf{E}} \subseteq \mathbf{E}$. That is, we want to find the model for which BP best approximates the marginals and the partition function of the original model. If one premises that, for comparison, we require the access to these exact quantities, the finding of such an edge set is of course an intractable problem. Still, it remains a crucial question whether and to what extent the removal of edges has a positive impact on the estimates. To identify edges to be deleted, we need to define an objective that contains information about the discrepancy between

<hr/>

[8]Without loss of generality, we assume that the removal of $\tilde{\mathbf{E}}$ does not make the graph disconnected (otherwise, individual connected components can be treated separately).

[9]Note that $\tilde{P}_\mathbf{X}'$ and $\tilde{Z}'$ are approximations to the exact marginals and partition function in the new model $(\mathcal{G}', \mathbf{\Phi}')$.

different graphical models. Note that a global comparison via the KL divergence and its generalizations [Minka, 2005] is prohibitive as this would involve a summation over exponentially many terms. Likewise, it is intractable to compare between different representations of the Gibbs free energy (5). To relax the problem, we focus on the analysis of local discrepancies between two models. The Bethe free energy (10) provides an ideal tool to explicitly measure these local differences.

## 3.2 THE BETHE ENERGY DIFFERENCE

Our main idea to make model comparison tractable lies in comparing between two different representations of the Bethe free energy. We formalize this concept as follows: assume for now that we remove a single edge $(i, j)$ from a model $(\mathcal{G}, \mathbf{\Phi})$ and let $(\mathcal{G}^{\setminus(i,j)}, \mathbf{\Phi}^{\setminus(i,j)})$ denote the resulting model. Let further $\mathcal{F}_B$ resp. $\mathcal{F}_B^{\setminus(i,j)}$ be the representations of the Bethe free energy that are associated with $(\mathcal{G}, \mathbf{\Phi})$ resp. $(\mathcal{G}^{\setminus(i,j)}, \mathbf{\Phi}^{\setminus(i,j)})$. Specifically, $\mathcal{F}_B^{\setminus(i,j)}$ does not contain the pairwise energy and entropy contributions from edge $(i, j)$, while the local entropy contributions from nodes $i$ and $j$ are counted once less than in the definition of $\mathcal{F}_B$ (10). Then we define the *Bethe free energy difference* $\Delta\mathcal{F}_B^{(i,j)}$ as the difference between $\mathcal{F}_B$ and $\mathcal{F}_B^{\setminus(i,j)}$, i.e.,

$$
\Delta\mathcal{F}_B^{(i,j)} := \mathcal{F}_B - \mathcal{F}_B^{\setminus(i,j)}
$$

$$
= \overbrace{-(1 + 2(2\xi_{ij} - q_i - q_j))J_{ij}}^{:=\Delta\mathcal{U}_B^{(i,j)}} \quad (18)
$$

$$
+ \underbrace{\mathcal{S}_i + \mathcal{S}_j - \mathcal{S}_{ij}}_{:=I_B^{(i,j)}},
$$

where $\Delta\mathcal{U}_B^{(i,j)}$ is the difference in the Bethe average energy and $I_B^{(i,j)}$ is the mutual information between $X_i$ and $X_j$.

Depending on whether we consider $\mathcal{F}_B$ on the local polytope $\mathbb{L}$ (13) or the Bethe box $\mathbb{B}$ (17), $\Delta\mathcal{F}_B^{(i,j)}$ is defined on slices of these objects, that is either on the sliced local polytope

$$
\mathbb{L}^{(i,j)} := \{(q_i, q_j; \xi_{ij}) \in \mathbb{R}^3 : 0 < q_i, q_j < 1; \\ \max(0, q_i + q_j - 1) < \xi_{ij} < \min(q_i, q_j)\} \quad (19)
$$

or the sliced Bethe box

$$
\mathbb{B}^{(i,j)} := \{(q_i, q_j) \in \mathbb{R}^2 : 0 < q_i, q_j < 1\}. \quad (20)
$$

It only depends on three resp. two variables and may therefore be considered as a function that contains variational information about local changes in a model when removing an edge. Moreover, it entails an effective way of measuring the local discrepancy between two graphical models, e.g., by computing an arbitrary norm of $\Delta\mathcal{F}_B^{(i,j)}$ on $\mathbb{L}^{(i,j)}$ or $\mathbb{B}^{(i,j)}$. In this work, we consider $L^p$-norms as the most

natural choice and analyze the special cases of $p = \infty$ and $p = 2$ in Sec. 3.3 (Theorem 1, Corollary 1, and Theorem 2).

In principle, one can generalize the above idea to compare between models that result from removing multiple edges $\tilde{\mathbf{E}}$ in one step, as the associated Bethe free energy difference $\Delta \mathcal{F}_B^{\tilde{\mathbf{E}}}$ is then simply the sum over energy differences $\Delta \mathcal{F}_B^{(i,j)}$ for all $(i,j)$ in $\tilde{\mathbf{E}}$. However, this increases both the number of variables to be integrated over and the number of edge sets to be taken into account for removal. To facilitate the theoretical and experimental analysis of edge removal, we shall therefore focus on removing edges one by one.

To make statements about the global effects of removing individual edges on the Bethe free energy and BP, we must carefully analyze the functional behavior of the Bethe free energy difference and its components on a small scale. In the following, we derive a series of auxiliary theorems where we consider the mathematical properties of $\xi_{ij}^*$ from (16) (Lemma 1 and 2), the Bethe mutual information $I_B^{(i,j)}$ (Lemma 3), and the Bethe energy difference $\Delta \mathcal{F}_B^{(i,j)}$ (Lemma 4, 5, and 6) on their joint domain, the sliced Bethe box $\mathbb{B}^{(i,j)}$. These results – besides being interesting in themselves – are rather of technical nature and will help us in proving our main results in Sec. 3.3. All proofs for Sec. 3.2 and 3.3 are contained in the Appendix A.

First, we compute values of $\xi_{ij}^*$ in the center point of the sliced Bethe box:

**Lemma 1.** *Let $(i, j)$ be an edge. In the center point $(0.5, 0.5)$ of the sliced Bethe box $\mathbb{B}^{(i,j)}$, the unique $\xi_{ij}^*$ that can be located at a stationary point of $\mathcal{F}_B$ has the form*

$$\xi_{ij}^*(0.5, 0.5) = \frac{\sigma(2\, J_{ij})}{2}. \tag{21}$$

We will also have to analyze the behavior of $\xi_{ij}^*$ if $q_i$ and $q_j$ approach the boundary $\partial \mathbb{B}^{(i,j)\,10}$ of the sliced Bethe box:

**Lemma 2.** *Let $(i, j)$ be edge and let $k \in [0, 1]$. The limits of $\xi_{ij}^*$ at the boundary $\partial \mathbb{B}^{(i,j)}$ of the sliced Bethe box are*

$$\lim_{\substack{q_i \to 0 \\ q_j \to k}} \xi_{ij}^*(q_i, q_j) = 0 = \lim_{\substack{q_i \to k \\ q_j \to 0}} \xi_{ij}^*(q_i, q_j), \tag{22}$$

$$\lim_{\substack{q_i \to 1 \\ q_j \to k}} \xi_{ij}^*(q_i, q_j) = k = \lim_{\substack{q_i \to k \\ q_j \to 1}} \xi_{ij}^*(q_i, q_j). \tag{23}$$

Moreover, we shall prepare bounds and compute the boundary limits at $\partial \mathbb{B}^{(i,j)}$ of the mutual information $I_B^{(i,j)}$ (18):

**Lemma 3.** *Let $(i, j)$ be an edge.*

(a) *In the interior of the sliced Bethe box $\mathbb{B}^{(i,j)}$, the mutual*

---

[10] That is, the four line segments connecting the points $(0, 0) - (0, 1)$, $(0, 0) - (1, 0)$, $(0, 1) - (1, 1)$, and $(1, 0) - (1, 1)$.

---

*information $I_B^{(i,j)}$ is bounded by*

$$\begin{aligned} 0 \;&<\; 8(\xi_{ij}^* - q_i q_j)^2 \\ &<\; I_B^{(i,j)}(q_i, q_j) \\ &\leq\; \frac{(\xi_{ij}^* - q_i q_j)^2}{q_i(1 - q_i) q_j(1 - q_j)}\,. \end{aligned} \tag{24}$$

(b) *The limit of $I_B^{(i,j)}$ at the boundary $\partial \mathbb{B}^{(i,j)}$ is*

$$\lim_{(q_i, q_j) \to \partial \mathbb{B}^{(i,j)}} I_B^{(i,j)}(q_i, q_j) = 0. \tag{25}$$

Next, we compute first-order and second-order derivatives of $\Delta \mathcal{F}_B^{(i,j)}$ on $\mathbb{B}^{(i,j)}$. The proof utilizes results from Welling and Teh [2001], Weller and Jebara [2013] (Appendix B).

**Lemma 4.** *Let $(i, j)$ be an edge.*

(a) *The first-order derivatives of $\Delta \mathcal{F}_B^{(i,j)}$ on $\mathbb{B}^{(i,j)}$ are*

$$\frac{\partial}{\partial q_i} \Delta \mathcal{F}_B^{(i,j)} = 2 J_{ij} + \log \left( \frac{(1 - q_i)(q_i - \xi_{ij}^*)}{q_i(1 + \xi_{ij}^* - q_i - q_j)} \right). \tag{26}$$

(b) *The second-order derivatives of $\Delta \mathcal{F}_B^{(i,j)}$ on $\mathbb{B}^{(i,j)}$ are*

$$\frac{\partial^2}{\partial q_i^2} \Delta \mathcal{F}_B^{(i,j)} = \frac{q_j(1 - q_j)}{T_{ij}} - \frac{1}{q_i(1 - q_i)}, \tag{27}$$

$$\frac{\partial^2}{\partial q_i q_j} \Delta \mathcal{F}_B^{(i,j)} = \frac{\partial^2}{\partial q_j q_i} \Delta \mathcal{F}_B^{(i,j)} = \frac{q_i q_j - \xi_{ij}^*}{T_{ij}}, \tag{28}$$

$$\frac{\partial^2}{\partial q_j^2} \Delta \mathcal{F}_B^{(i,j)} = \frac{q_i(1 - q_i)}{T_{ij}} - \frac{1}{q_j(1 - q_j)}, \tag{29}$$

*where $T_{ij} := q_i q_j(1 - q_i)(1 - q_j) - (\xi_{ij}^* - q_i q_j)^2$.*

The following result formulates a useful property of the Bethe free energy difference on the sliced Bethe box:

**Lemma 5.** *Let $(i, j)$ be an edge. $\Delta \mathcal{F}_B^{(i,j)}$ has precisely one stationary point on $\mathbb{B}^{(i,j)}$, which is $(\bar{q}_i, \bar{q}_j) = (0.5, 0.5)$ and is neither a maximum nor a minimum (i.e., a saddle point).*

Lemma 5 implies that $\Delta \mathcal{F}_B^{(i,j)}$ cannot possess a maximum nor a minimum in the interior of $\mathbb{B}^{(i,j)}$. This implies that the supremum and infimum of $\Delta \mathcal{F}_B^{(i,j)}$ must lie at the boundary. Finally, we characterize regions of $\mathbb{B}^{(i,j)}$ on which $\Delta \mathcal{F}_B^{(i,j)}$ contributes always negatively to the Bethe free energy $\mathcal{F}_B$:

**Lemma 6.** *Let $(i, j)$ be an edge.*

(a) *For an attractive edge, $\Delta \mathcal{F}_B^{(i,j)}(q_i, q_j)$ is negative if either both $q_i, q_i < 0.5$ or both $q_i, q_i > 0.5$.*

(b) *For a repulsive edge, $\Delta \mathcal{F}_B^{(i,j)}(q_i, q_j)$ is negative if either $q_i < 0.5$ and $q_j > 0.5$ or $q_i < 0.5$ and $q_j > 0.5$.*

### 3.3 MAIN RESULTS

After having prepared the technical framework in Sec. 3.2, we now proceed by presenting our main results. First, we directly relate the Bethe free energy difference to the local properties of the graphical model (Theorem 1). Then, we address the problem of an 'Bethe-optimal' edge to be deleted (Corollary 1 and Theorem 2). Finally, we conclude about the approximation quality of BP regarding the estimated partition function if edges are deleted (Theorems 3 and 4).

**Theorem 1.** *Let $(i, j)$ be an arbitrary edge. Then the $L^\infty$-norm of the Bethe free energy difference is*

$$||\Delta \mathcal{F}_B^{(i,j)}||_{L^\infty} = |J_{ij}|, \qquad (30)$$

*with*

$$-|J_{ij}| = \inf_{(q_i, q_j) \in \mathbb{B}^{(i,j)}} \Delta \mathcal{F}_B^{(i,j)}(q_i, q_j), \qquad (31)$$

$$|J_{ij}| = \sup_{(q_i, q_j) \in \mathbb{B}^{(i,j)}} \Delta \mathcal{F}_B^{(i,j)}(q_i, q_j). \qquad (32)$$

*The infimum and supremum are not taken by $\Delta \mathcal{F}_B^{(i,j)}$ but exist only as limits at the boundary of $\mathbb{B}^{(i,j)}$. In particular, we have for an attractive edge*

$$-J_{ij} = \lim_{\substack{q_i \to 0 \\ q_j \to 0}} \Delta \mathcal{F}_B^{(i,j)}(q_i, q_j) = \lim_{\substack{q_i \to 1 \\ q_j \to 1}} \Delta \mathcal{F}_B^{(i,j)}(q_i, q_j),$$

$$J_{ij} = \lim_{\substack{q_i \to 0 \\ q_j \to 1}} \Delta \mathcal{F}_B^{(i,j)}(q_i, q_j) = \lim_{\substack{q_i \to 1 \\ q_j \to 0}} \Delta \mathcal{F}_B^{(i,j)}(q_i, q_j),$$

$$(33)$$

*and, conversely, for a repulsive edge*

$$-J_{ij} = \lim_{\substack{q_i \to 0 \\ q_j \to 1}} \Delta \mathcal{F}_B^{(i,j)}(q_i, q_j) = \lim_{\substack{q_i \to 1 \\ q_j \to 0}} \Delta \mathcal{F}_B^{(i,j)}(q_i, q_j),$$

$$J_{ij} = \lim_{\substack{q_i \to 0 \\ q_j \to 0}} \Delta \mathcal{F}_B^{(i,j)}(q_i, q_j) = \lim_{\substack{q_i \to 1 \\ q_j \to 1}} \Delta \mathcal{F}_B^{(i,j)}(q_i, q_j).$$

$$(34)$$

Theorem 1 reveals a monotonic dependence between the strength of the couplings and absolute changes in the Bethe free energy that are caused by local modifications in the graphical structure. In terms of edge deletion, this implies that the 'Bethe-optimal' choice of an edge to be removed from the graph is the one with the weakest coupling strength:

**Corollary 1.** *Suppose we aim to remove an edge from the graphical model such that the induced maximum error in the Bethe free energy is minimal. Then this '$L^\infty$-Bethe-optimal' edge is the one with the lowest absolute coupling strength:*

$$\operatorname*{argmin}_{(i,j) \in \mathbf{E}} ||\Delta \mathcal{F}_B^{(i,j)}||_{L^\infty} = \operatorname*{argmin}_{(i,j) \in \mathbf{E}} |J_{ij}| \qquad (35)$$

An analogous property holds for the $L^2$-error of $\mathcal{F}_B$ on $\mathbb{L}$:

**Theorem 2.** *Suppose we aim to remove an edge from the graphical model such that the induced mean squared error in the Bethe free energy on the local polytope $\mathbb{L}$ is minimal. Then this '$L^2$-Bethe-optimal' edge is the one with the lowest absolute coupling strength:*

$$\operatorname*{argmin}_{(i,j) \in \mathbf{E}} ||\Delta \mathcal{F}_B^{(i,j)}||_{L^2} = \operatorname*{argmin}_{(i,j) \in \mathbf{E}} |J_{ij}| \qquad (36)$$

Next, we conclude about the quantitative change in the Bethe partition function $\mathcal{Z}_\mathcal{B}$ if an edge is removed:

**Theorem 3.** *Let $\mathcal{Z}_\mathcal{B}$ be the Bethe partition function associated with some graphical model, i.e., the quanitity that satisfies $-\log(\mathcal{Z}_\mathcal{B}) = \min_\mathbb{B} \mathcal{F}_B$. Suppose we remove an (attractive or repulsive) edge from the graph. Let $\mathcal{F}_B^{\backslash(i,j)}$ be the representation of the Bethe free energy associated with the new model, together with the new Bethe partition function $\mathcal{Z}_\mathcal{B}^{\backslash(i,j)}$ that is implicitly defined by $-\log(\mathcal{Z}_\mathcal{B}^{\backslash(i,j)}) = \min_\mathbb{B} \mathcal{F}_B^{\backslash(i,j)}$. Then the following error estimate holds:*

$$\left| \log \left( \frac{\mathcal{Z}_\mathcal{B}}{\mathcal{Z}_\mathcal{B}^{\backslash(i,j)}} \right) \right| < |J_{ij}| \qquad (37)$$

Finally, we conclude about the quality of the estimated partition function if edges are removed. We consider unidirectional models (Sec. 2.1) that allow for a precise statement:

**Theorem 4.** *Consider a unidirectional model, i.e., where all edges are attractive and all variables are biased towards the same state. Let $\mathcal{Z}$ resp. $\mathcal{Z}_\mathcal{B}$ be the associated partition resp. Bethe partition function. Suppose we remove an arbitrary edge from the graph and let $\mathcal{Z}_\mathcal{B}^{\backslash(i,j)}$ be the Bethe partition function associated with the new model. Then the quality of the estimated partition function degrades, i.e.,*

$$|\mathcal{Z} - \mathcal{Z}_\mathcal{B}| < |\mathcal{Z} - \mathcal{Z}_\mathcal{B}^{\backslash(i,j)}|. \qquad (38)$$

Theorem 4 does not formally extend to models that contain both positive and negative local fields. Generally, however, the error between the true and the BP-estimated partition function tends to increase, the more edges we remove.

This negative result is contrasted by the positive effect of edge removal on the estimated marginals. While existing theoretical bounds on the marginal errors are often loose and typically hard to compute [Wainwright et al., 2003, Taga and Mase, 2006, Ihler, 2007, Mooij and Kappen, 2008], we validate and explain our statement in Sec. 4.

## 4 EXPERIMENTS

We now demonstrate empirically how removing edges can have an astonishingly positive impact on the approximation accuracy of BP. We perform a range of experiments on a

fully connected graph on 10 vertices.[11] Further experiments including a $5 \times 5$- grid graph are contained in Appendix C.

We consider both attractive and general models (Sec. 2.1). In Sec. 4.1, we focus on attractive models and sample $J_{ij}$ uniformly from $[0, \hat{J}]$ for $\hat{J} \in \{0.1, 0.2, \ldots, 2\}$. In Sec. 4.2, we focus on general models and sample $J_{ij}$ uniformly from $[-\hat{J}, \hat{J}]$ for $\hat{J} \in \{0.1, 0.2, \ldots, 2\}$. For both settings, we create two scenarios: first, models with weak local fields (each $\theta_i$ is sampled uniformly from $[-0.2, 0.2]$); second, models with strong local fields (each $\theta_i$ is sampled uniformly from $[-0.5, 0.5]$). For each configuration, we create 200 models.

For each individual model, we remove edges one by one until we reach a spanning tree. We do not remove edges, whose deletion makes the graph disconnected.[12] We compare two criteria for selecting the next edge to be removed: first, the Bethe-optimal criterion (Corollary 1, Theorem 2); second, we remove edges that induce the lowest mutual information between two connected variables in the original model. More precisely: assume that we have already removed edge set $\tilde{\mathbf{E}}$ from a model; then the next edge $(i, j)$ to be removed is the one that minimizes either of the following criteria:

$$\text{BETHE-OPT}: \quad \operatorname*{argmin}_{(i,j) \in \mathbf{E} \setminus \tilde{\mathbf{E}}} |J_{ij}|$$

$$\text{CHOW-LIU}: \quad \operatorname*{argmin}_{(i,j) \in \mathbf{E} \setminus \tilde{\mathbf{E}}} I(X_i; X_j)$$

Note that by applying the second criterion, we end up in a Chow-Liu tree [Chow and Liu, 1968], i.e., the spanning tree with the lowest KL divergence from the original model.[13]

For each intermediate model during the edge deletion process, we run BP 100 times with random message initialization to approximate the marginals. For each run, we perform at most 1000 iterations. If BP has not converged, we estimate the marginals from the final iteration. We utilize a randomized message scheduling to achieve better convergence [Elidan et al., 2006]. For the error evaluation, we compute the $l^1$-distance between the exact and estimated marginals. The results for each model are averaged over the 100 runs. Finally, the results are averaged over all 200 models, each based on a different configuration of the potentials.

## 4.1   ATTRACTIVE MODELS

For weak couplings, BP finds accurate marginal estimates in the original model. If the strength of the couplings increases,

this favorable property suddenly disappears at some critical treshold and BP fails to approximate the marginals for larger values of $\hat{J}$ (Fig. 1). This behavior is not due to worse convergence properties of BP, but results from inaccurate BP fixed points and thus inaccurate minima of the Bethe free energy [Weller et al., 2014]. While the Bethe free energy is convex for weaker couplings and possesses a unique global minimum, this minimum becomes an (unstable) saddle point if $\hat{J}$ increases and cannot be reached by BP any longer [Heskes, 2003, Mooij and Kappen, 2005, Knoll and Pernkopf, 2017]. For even larger couplings, the landscape of the Bethe free energy becomes increasingly complex and the (possibly many[14]) Bethe minima approach the boundary of the domain, thus moving away from the exact marginals.

If we remove edges from the graph, the marginal accuracy of BP in the new model is often much better than in the original model. This can be explained by a 'reconvexification' of the Bethe free energy that makes unstable saddle points or maxima stable minima again and allows BP to converge to accurate fixed points. The question on how many edges we should actually remove, is a difficult one. In Fig. 1, we observe that there appears to be a 'channel' that defines an optimal number of edges to be removed. The stronger the couplings become, the more preferable is it to rely on tree approximations, while BP outperforms the edge removal techniques for regimes with lower coupling strength (Fig. 2). For stronger local potentials, the channel becomes narrower and edge removal loses some of its benefit (although the results are mostly superior in comparison to the original model). Also, in Fig. 1 we observe that BETHE-OPT performs slightly better than CHOW-LIU criterion, with an increasing advantage for stronger local potentials.

## 4.2   GENERAL MODELS

The situation for general models is similar as in the attractice case. We can observe certain differences though (Fig. 3): first, the critical treshold of the couplings, beyond which the Bethe free energy becomes non-convex, is higher than for attractive models. Second, for models with strong local potentials, edge removal based on the BETHE-OPT criterion improves the marginal accuracy only slightly. Interestingly, the Chow-Liu tree induces strikingly accurate Bethe minima for all models (Fig. 4). As in the attractive case, we observe that the problem becomes more difficult if both the pairwise and the local potentials become stronger at the same time.

## 5   CONCLUSION

We have proposed to approximate a graphical model as a 'preprocessing step' for approximate inference. We focused

---

[11]This allows for a computation of the exact marginals via the junction tree algorithm [Lauritzen and Spiegelhalter, 1988] and enables us to compare the approximated marginals to them.

[12]This procedure corresponds to the so-called reverse-delete algorithm [Kruskal, 1956] that constructs a maximum spanning tree with respect to a given criterion.

[13]We cannot generally apply the second criterion, as the computation of the mutual information between two variables requires knowledge of the related exact singleton and pairwise marginals.

---

[14]The Bethe free energy may theoretically possess exponentially many local minima [Watanabe and Fukumizu, 2009, Knoll and Pernkopf, 2019].

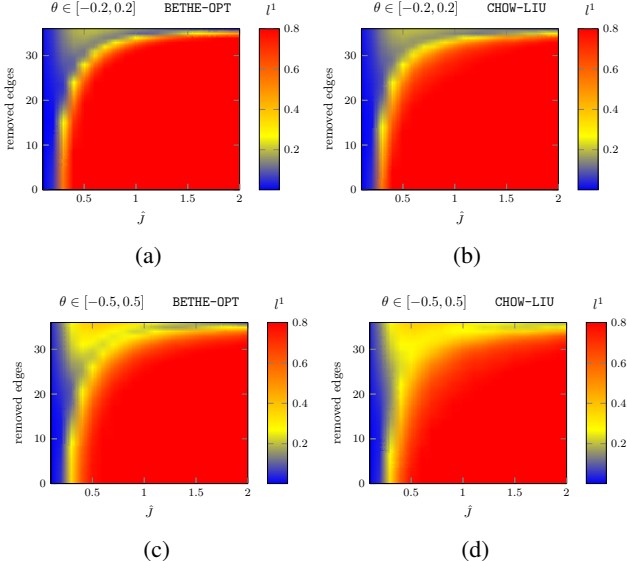

(a)  (b)

(c)  (d)

Figure 1: Attractive models. First row: $\theta_i \in [-0.2, 0.2]$; second row: $\theta_i \in [-0.5, 0.5]$. (a) + (c): BETHE-OPT criterion; (b) + (d): CHOW LIU criterion.

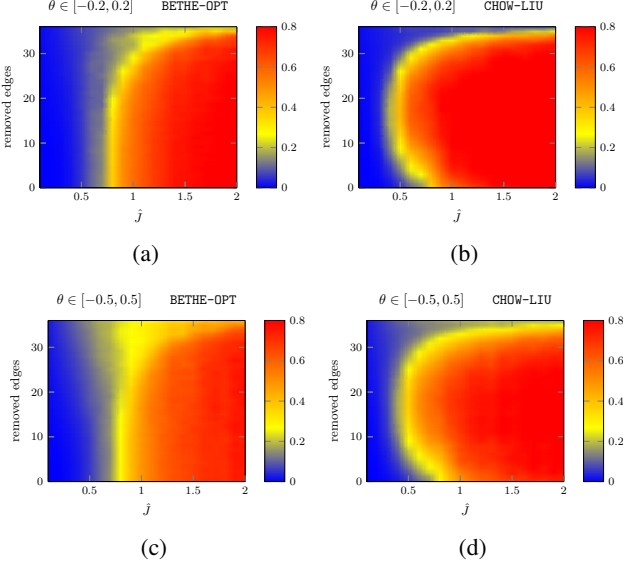

(a)  (b)

(c)  (d)

Figure 3: General models. First row: $\theta_i \in [-0.2, 0.2]$; second row: $\theta_i \in [-0.5, 0.5]$. (a) + (c): BETHE-OPT criterion; (b) + (d): CHOW LIU criterion.

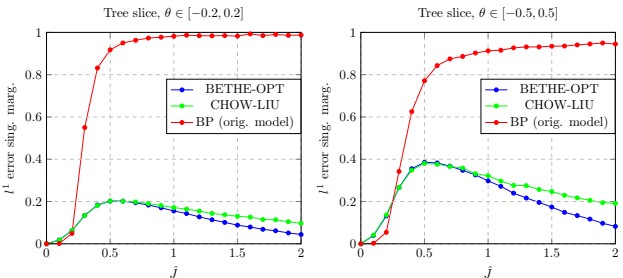

Figure 2: Attractive models. Tree approximations with respect to BETHE-OPT (blue) and CHOW-LIU (green) vs. BP in the original model (red). Left-hand side: $\theta_i \in [-0.2, 0.2]$; right-hand side: $\theta_i \in [-0.5, 0.5]$.

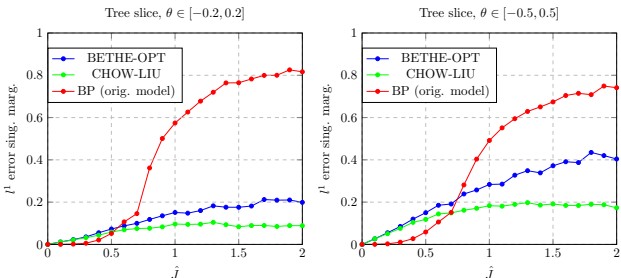

Figure 4: General models. Tree approximations with respect to BETHE-OPT (blue) and CHOW-LIU (green) vs. BP in the original model (red). Left-hand side: $\theta_i \in [-0.2, 0.2]$; right-hand side: $\theta_i \in [-0.5, 0.5]$.

**Acknowledgements**

This work was supported by the Graz University of Technology LEAD project "Dependable Internet of Things in Adverse Environments". We further acknowledge support from the Wireless Lab, Huawei Technologies Sweden.

on the removal of single edges and showed that this can have a beneficial impact on the behavior of belief propagation.

We have exploited the relationship between belief propagation and the Bethe free energy to explain the success of such an approach. Subsequently, we have validated our findings in an experimental study. Most importantly, our analysis contributes to an improved understanding of belief propagation and the Bethe approximation in general.

We are convinced that our observations inspire the development of further sophisticated methods that try to approximate a graphical model and improve the behavior of message passing algorithms. We believe that one logical extension lies in the modification of the local potentials to compensate for the lost information caused by edge removal.

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
