# OpenReview forum: "Fixing the Bethe Approximation: How Structural Modifications in a Graph Improve Belief Propagation"
_auai.org/UAI/2022/Conference — UAI 2022 Oral_

### Official Review · Reviewer_zP7a · 2022-04-13

**Q2(1) Originality/Novelty:** 3
**Q2(2) Significance/Impact:** 4
**Q2(3) Correctness/Technical Quality:** 3
**Q2(6) Clarity Of Writing:** 4
**Q6 Overall Score:** 8
**Q8 Confidence In Your Score:** 3

**Q1 Summary And Contributions:**

This paper proves several results relating augmentation of an undirected graphical mode to changes in the resulting Bethe free energy approximation to the partition function and demonstrates empirically that approximation of marginals in the graph can actually improve through these augmentations.

**Q2 Assessment Of The Paper:**

More detailed information regarding each of these aspects is given below:

**Q2(4) Quality Of Experiments (Optional):**

3: Good: The experimental evaluation is adequate, and the results convincingly support the main claims.

**Q2(5) Reproducibility:**

4: Excellent: Key resources (e.g., proofs, code, data) are available and key details (e.g., proof sketches, experimental setup) are comprehensively described for competent researchers to confidently and easily reproduce the main results.

**Q3 Main Strengths:**

See below.

**Q4 Main Weakness:**

See below.

**Q5 Detailed Comments To The Authors:**

I found this to be and excellently written paper that provided interesting insights into a heavily studied algorithm. The coverage of background and related work was comprehensive and clear, the presentation of results was concise and intuitive, and the experiments were very well described and provided genuine insight into the behavior of the proposed method. I don't have much negative to say about this paper.

One question that comes to mind is how a method like this affects parameter estimation. Specifically, one potential goal of approximating the marginals is to approximate the gradient of the log-likelihood. Does this approach to approximation (i.e., removing edges from the graph and using loopy-BP) have any implications for model estimation using the approximated marginals to approximate gradients?

Minor comments and typos:

1. I would recommend adding a bit of discussion or context around the lemmas in Section 3.2. It was not totally clear to me what purpose the serve beyond use in the proof of later results.

2. Page five, last paragraph: proves --> proofs

**Q7 Justification For Your Score:**

Excellently written paper with novel and interesting insights into a long-standing and important inference algorithm.

**Q9 Complying With Reviewing Instructions:**

1: Yes.

---

### Official Review · Reviewer_J6fX · 2022-04-13

**Q2(1) Originality/Novelty:** 3
**Q2(2) Significance/Impact:** 3
**Q2(3) Correctness/Technical Quality:** 3
**Q2(6) Clarity Of Writing:** 3
**Q6 Overall Score:** 6
**Q8 Confidence In Your Score:** 3

**Q1 Summary And Contributions:**

Loopy belief propagation often has faster convergence and better inference performance than structured mean field. However theoretical analysis is often difficult. It is good to see a serious piece of theoretical work on Bethe free energy. While I do not have time to check all the proofs, it seems that the background review is accurate and the problem to address is an important one.

**Q2 Assessment Of The Paper:**

More detailed information regarding each of these aspects is given below:

**Q2(4) Quality Of Experiments (Optional):**

3: Good: The experimental evaluation is adequate, and the results convincingly support the main claims.

**Q2(5) Reproducibility:**

3: Good: Key resources (e.g., proofs, code, data) are available and key details (e.g., proofs, experimental setup) are sufficiently well-described for competent researchers to confidently reproduce the main results.

**Q3 Main Strengths:**

The authors seem to have address an important theoretical problem on Bethe variational principle. The literature review is accurate. And the approach is likely to work.

**Q4 Main Weakness:**

I have some doubt about Eq. 30. While removing an edge removes some local constraints, it also changes all pseudo-marginals and the contributions of all pseudo marginals to the overall Bethe variational principle. I do not remember there is a simple characterization of this, and find that taking marginals $q_i$ to approach 0 or 1 suspicious. Any intuition that help me with saying that the proof is likely correct or fixable without affording a huge amount of time to check the proofs?

**Q5 Detailed Comments To The Authors:**

I like this serious piece of theoretical work. However I haven't been working with Bethe free energy for a long time and cannot afford the amount of time to thoroughly check all the proofs. Any intuition that help me with saying that the proof of theorem 1 is likely correct or fixable without affording a huge amount of time to check the proofs?

**Q7 Justification For Your Score:**

The problem statement and the literature review is accurate and comprehensive. And the proposed approach is likely to work. However, behind the long constructive proofs I suspect that some important assumptions are missing (so the achievement might be exaggerated).

**Q9 Complying With Reviewing Instructions:**

1: Yes.

---

### Official Review · Reviewer_dMqW · 2022-04-14

**Q2(1) Originality/Novelty:** 3
**Q2(2) Significance/Impact:** 2
**Q2(3) Correctness/Technical Quality:** 3
**Q2(6) Clarity Of Writing:** 4
**Q6 Overall Score:** 6
**Q8 Confidence In Your Score:** 4

**Q1 Summary And Contributions:**

The paper studies how removing edges from a graphical model influences the properties of Belief propagation (BP). The research is motivated by the high computational complexity of reasoning in probabilistic graphical models and the main contribution are greedy-based optimal methods to select an edge to be removed from the graph.

**Q2 Assessment Of The Paper:**

More detailed information regarding each of these aspects is given below:

**Q2(4) Quality Of Experiments (Optional):**

2: Fair: The experimental evaluation is weak: important baselines are missing, or the results do not adequately support the main claims.

**Q2(5) Reproducibility:**

3: Good: Key resources (e.g., proofs, code, data) are available and key details (e.g., proofs, experimental setup) are sufficiently well-described for competent researchers to confidently reproduce the main results.

**Q3 Main Strengths:**

- The authors investigate an important problem how, for a given graphical model, an edge removal influences the marginals and the estimated partition function and they provide an effective greedy-based edge deletion strategy which is Bethe-optimal.
- The results are not very surprising, but the analysis is quite involved and highly non-trivial.
- Well-motivated problem studied both theoretically and experimentally.
- Technically solid and well-written paper.

**Q4 Main Weakness:**

- The results of the experimental analysis presented in Section 4 are not very convincing.

- More precisely, the authors compare the performance of the proposed method, called here BETHE-OPT with the Chow-Liu algorithm [Chow and Liu, 1968] which finds a spanning tree such that the Kullback-Leibler divergence between the original distribution and the tree distribution is minimal. In Fig. 2 we see that, in fact, for the intermediate models during the edge deletion process,  BETHE-OPT outperforms the Chow-Liu. However when the algorithms reach a spanning tree then the model computed by the Chow-Liu algorithm has significantly smaller error that the model computed by BETHE-OPT.

- Though the edge-deletion significantly improves the marginal accuracy of BP,  the quality of the estimated partition function tends to deviate from the original model, which is a big disadvantage of this approach.


**Q5 Detailed Comments To The Authors:**

All theoretical results are done for L2 and L-infty distance but in the experimental analysis L1 is used.

P.5: You write: Ideally, we would like to remove a set of edges, such that the induced errors become minimal over all subsets ~E ⊆ E. Note that you should exclude the empty set.

P.5: You write that to find the model for which BP best approximates the marginals and the partition function of the original model  is an intractable problem. Could you justify this?

P.5: You argue: To facilitate the theoretical and experimental analysis of edge removal, we shall therefore focus on removing edges one by one. What about other strategies?

Lemma 1. Explain the meaning of σ in the main paper.

Thm. 1: the meaning of |J_ij | as  explained in Eq. (31) and (32) is not clear.

P.3:  are the local entropies  ->  and the local entropies are


**Q7 Justification For Your Score:**

Technically solid paper presenting interesting theoretical analysis. However, the results of the experimental analysis are not very convincing and thus, the importance of the main results is not clear.

**Q9 Complying With Reviewing Instructions:**

1: Yes.

---

### Decision · Program_Chairs · 2022-05-15

**Decision:**

Accept (Oral)

**Comment:**

Meta Review: All reviewers appreciated the theoretical contributions being made towards understanding BP and the Bethe approximation.